# A Near-Infrared Fluorescent Probe for Recognition of Hypochlorite Anions Based on Dicyanoisophorone Skeleton

**DOI:** 10.3390/molecules28010402

**Published:** 2023-01-03

**Authors:** Chang-Xiang Liu, Shu-Yuan Xiao, Xiu-Lin Gong, Xi Zhu, Ya-Wen Wang, Yu Peng

**Affiliations:** 1School of Life Science and Engineering, Southwest Jiaotong University, Chengdu 610031, China; 2Department of Neurology, The Third People’s Hospital of Chengdu, The Affiliated Hospital of Southwest Jiaotong University, Chengdu 610031, China

**Keywords:** fluorescence, hypochlorite anion (ClO^−^), near-infrared probe, chemodosimeter

## Abstract

A novel near-infrared (NIR) fluorescent probe (**SWJT-9**) was designed and synthesized for the detection of hypochlorite anion (ClO^−^) using a diaminomaleonitrile group as the recognition site. **SWJT-9** had large Stokes shift (237 nm) and showed an excellent NIR fluorescence response to ClO^−^ with the color change under the visible light. It showed a low detection limit (24.7 nM), high selectivity, and rapid detection (within 2 min) for ClO^−^. The new detection mechanism of **SWJT-9** on ClO^−^ was confirmed by ^1^H NMR, MS spectrum, and the density functional theory (DFT) calculations. In addition, the probe was successfully used to detect ClO^−^ in HeLa cells.

## 1. Introduction

Neutrophils, known as polymorphonuclear cells, are the largest number of white blood cells in the body [1,2,3]. They are the main immune cells that protect the body from microbial infection and eliminate pathogens [4]. In neutrophils, hydrogen peroxide (H_2_O_2_) reacts with chloride ions to generate hypochlorite anions [5,6,7] under the catalysis of myeloperoxidase. Hypochlorite anions plays a very important role in the human body [8,9]. However, excessive hypochlorite anion in the body will oxidize biological molecules, such as protein, cholesterol, DNA, and RNA in living cells, which will lead to cardiovascular disease, inflammatory disease, cancer, and so on [10,11,12,13,14,15]. Therefore, it is necessary to monitor hypochlorite anions in vitro and in vivo. 

At present, many analytical methods have been applied to the detection of hypochlorite anions, such as colorimetry, luminescence, electrochemistry, and chromatography [16,17,18]. In addition, the fluorescence probing method was paid more attention as an excellent detection tool, which has realized the detection of many active species [19,20,21,22,23]. Fluorescence probes for the detection of hypochlorite anions have been reported continuously in recent years [24,25,26,27,28,29,30,31,32,33,34,35,36,37,38,39,40,41,42,43,44,45,46]. Among them, some chemodosimeters using a diaminomaleonitrile group as the reaction site were used to detect hypochlorite anions, which had many advantages, such as specific recognition, high selectivity, and fast response. However, most chemodosimeters for the hypochlorite anion could hardly achieve near-infrared fluorescence emission and did not have a large Stokes shift (Appendix A) [34,35,36,37,38,39,40,41,42,43,44,45,46]. It is well known that NIR fluorescent probes with a large Stokes shift have more advantages due to good applications in biological systems [47]. Therefore, it is necessary to design a near-infrared fluorescent probe with a large Stokes shift for the detection of hypochlorite anions with high selectivity and rapid response. 

In connection with our previous work [26,48,49], herein, we reported a novel chemodosimeter (**SWJT-9**), which had a large Stokes shift and emitted near-infrared fluorescence due to using a dicyanoisophorone skeleton. It can specifically recognize a hypochlorite anion based on the inhibition of C=N rotational isomerization [50]. Moreover, **SWJT-9** was successfully applied to HeLa cell imaging. 

## 2. Results and Discussion

### 2.1. Design **SWJT-9**

Through the Duff reaction, an aldehyde group was generated at the ortho position of the hydroxyl group in a dicyanoisophorone skeleton. The diaminomaleonitrile group was then used as the recognition group [36,51] to obtain the probe **SWJT-9** (Figure 1). The structure of **SWJT-9** was confirmed by NMR and MS (Appendix A). The fluorescence of **SWJT-9** would reduce by the rotational isomerization of the C=N double bond [52]. After the addition of hypochlorite anions, a reaction between **SWJT-9** and ClO^−^ would occur to obtain compound **3**, which had no C=N bond; therefore, the fluorescence would be enhanced [53,54,55].

### 2.2. Photoproperties of Probe

In order to study the effect of organic solvents on the probe, methanol, ethanol, acetonitrile, DMSO, and DMF were used (Appendix A). The effect of the buffer solution and pH on the probe were also studied (Appendix A). According to these results, and considering the solubility of the probe, the solution of ethanol and PBS (9/1, *v*/*v*) at pH 7.4 was selected as the test condition.

As shown in Figure 1a, the UV–Vis absorption spectrum of **SWJT-9** showed an obvious absorption band centered at 430 nm. After the addition of ClO^−^ to the solution, the absorbance red-shifted to about 550 nm. The color of the solution changed from yellow to pink (Figure 1a, inset). These results suggested that a reaction might occur between **SWJT-9** and ClO^−^. In the fluorescence spectrum, **SWJT-9** showed weak emission at about 667 nm (Φ = 0.018) under excitation at 550 nm (Figure 1b). After the addition of ClO^−^, the fluorescence increased (Φ = 0.113) [56], and the fluorescence color of the solution was observed from red to deep red (Figure 1b, inset). These results indicated that **SWJT-9** could detect hypochlorite anions by colorimetric and fluorescence turn-on responses. The fluorescence titration experiments were then conducted. As shown in Figure 1c, the fluorescence intensity enhanced with the increase of the ClO^−^ concentration, indicating that **SWJT-9** was a turn-on fluorescence probe. The detection limit was calculated as 24.7 nM (Appendix A), which was far lower than the concentration of hypochlorite anions produced by cells [57]. 

In addition, it is well known that the reaction time is an important parameter for examining intracellular hypochlorite anions [28]. The shorter the time, the better the recognition. As shown in Figure 1d, the fluorescence intensity tends to equilibrate within two minutes. These results showed that this probe has high sensitivity to hypochlorite anions. Moreover, the constant (*k*_obs_) of the pseudo-first-order reaction was calculated to be 0.03041 s^−1^, and the *t*_1/2_ was 23 s (Appendix A). 

### 2.3. Competition Experiments

As show in Figure 2, when active oxygen species or other anions were added to the solution of **SWJT-9**, the fluorescence intensity was still weak. The fluorescence intensity at 667 nm significantly enhanced when ClO^−^ was added to the above solution. These results indicated that the presence of other reactive oxygen species or anions would not interfere with the recognition of ClO^−^ by **SWJT-9**. The probe had a good anti-interference property and potential application in biological environments.

### 2.4. Response Mechanism

In order to verify the possible reaction mechanism of **SWJT-9** and ClO^–^, ^1^H NMR titration (Figure 3) and MS spectrum (Appendix A) were used. As shown in Figure 3, the proton signal of the C=N bond (H_a_) appeared at 8.5 ppm. When ClO^–^ was added, the peak (H_a_) disappeared gradually while a new proton (H_b_) appeared. Compared with two spectrum (**SWJT-9** + ClO^−^ and compound **3**), they were basically the same, which indicated **3** was the product from a reaction of **SWJT-9** and ClO^−^. In addition, as shown in Appendix A, the peak at *m*/*z* 349.3 of **SWJT-9** + ClO^−^ corresponds to 3.

### 2.5. DFT Calculations

In order to investigate the relationship between the probe and the spectral changes, density functional theory (DFT, B3LYP/6-311G, Gaussian 09) calculations were performed [58]. As shown in Figure 4, the HOMO electron density of **SWJT-9** was distributed on the dicyanoisophorone skeleton and diaminomaleonitrile group. However, for the LUMO level, the electron density was only located on the dicyanoisophorone skeleton, which meant that C=N isomerization occurred to lead to the weak fluorescence of **SWJT-9** [59]. At the HOMO and LUMO levels, the electrons of compound **3** were all mainly distributed in the whole dicyanoisophorone group, indicating that compound **3** had a strong fluorescence emission. These results suggested that **SWJT-9** could be considered as a turn-on probe for ClO^−^.

### 2.6. Cytotoxicity of **SWJT-9** and Its Imaging in HeLa Cells

In order to further prove the excellent performance of the near-infrared fluorescence of **SWJT-9**, HeLa cells were used (Figure 5) to study bioimaging. First, HeLa cells were incubated with **SWJT-9** (10.0 μM) for 30 min. As shown in Figure 5b, the fluorescence of **SWJT-9** was very weak in the red channel. In the other group, **SWJT-9** and HeLa cells were incubated at 37 °C for 30 min and then incubated with ClO^–^ for another 20 min. The fluorescence in the red channel improved obviously. These results indicated that **SWJT-9** has good cell membrane transparency and can detect ClO^−^ in HeLa cells. In addition, the CCK test results showed that **SWJT-9** did not produce significant cytotoxicity in HeLa cells (Figure 6).

## 3. Materials and Methods 

### 3.1. Materials and Reagents

The materials, reagents, and detection methods are described in the Appendix A.

### 3.2. Synthesis of Probe SWJT-9

Compound **1** and compound **2** were prepared according to the previously reported literature [60].

Compound **2** (5.01 g, 15.61 mmol) and hexamethylenetetramine (4.16 g, 29.65 mmol) were dissolved in trifluoroacetic acid (20 mL), heated to 100 °C, and reacted for 8 h. The mixture solution was cooled to room temperature and extracted with CH_2_Cl_2_ (2 × 150 mL). Finally, the collected organic layers were concentrated and purified by column chromatography (petroleum ether:ethyl acetate = 4:1) on silica gel to obtain the known compound **3** [49] (1.80 g, 33.1%). 

Compound **3** (90.10 mg, 0.26 mmol) and diaminomaleonitrile (32.01 mg, 0.30 mmol) were added to ethanol (10 mL). Then, the mixture was heated at 80 °C for two hours and then filtered with suction. After washing with ethanol, **SWJT-9** can be obtained as a solid. Yield: 45.0 %. ^1^H NMR (400 MHz, DMSO-*d*_6_): *δ* = 10.34 (s, 1H), 8.58 (s, 1H), 8. 0 (s, 1H), 7.95 (s, 1H), 7.47 (s, 1H), 7.37 (d, *J* = 16.0 Hz, 1H), 7.22 (d, *J* = 16.0 Hz, 1H), 6.84 (s, 1H), 3.93 (s, 3H), 2. 62 (s, 2H), 2. 53 (s, 2H), 1. 03 (s, 6H) ppm. ^13^C NMR (100 MHz, DMSO-*d*_6_): *δ* = 170.7, 156.6, 151.8, 150.0, 148.9, 138.3, 128.0, 127.9, 126.9, 122.3, 122.3, 121.4, 114.9, 114.5, 114.3, 113.7, 113.2, 103.8, 75.9, 56.7, 42.8, 38.8, 32.2, 27.9 (2C) ppm. ESI-MS: *m*/*z* 439.1 [M + H]^+^. 

## 4. Conclusions

In conclusion, a near-infrared fluorescent probe **SWJT-9** for ClO^−^ detection was developed. It exhibited a large Stokes shift (237 nm), high selectivity, good sensitivity, and fast response. An obvious color change was observed by the naked eye under visible light or ultraviolet light. **SWJT-9** was a new near-infrared fluorescent probe using diaminomaleonitrile as the recognition group, and it can be successfully used in cells, indicating its potential use in biological analyses. This work could provide some inspiration for future research of near-infrared fluorescence probes.

## Data Availability

Not applicable.

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
