# Peer review of "A Near-Infrared Fluorescent Probe for Recognition of Hypochlorite Anions Based on Dicyanoisophorone Skeleton"

_molecules, 2023, doi:10.3390/molecules28010402_

Round 1

Author Response

Reply to Referee 1:Comments to the Author

In this manuscript, the authors reported a novel near-infrared (NIR) fluorescent probe (SWJT-9) for the detection of hypochlorite anion (ClO-) using a diaminomaleonitrile group as the recognition site. The probe has the advantages of large Stokes shift (237 nm), low detection limit (24.7 nM), high selectivity and rapid detection (within 2 min). In addition, probe SWJT-9 could be used to detect ClO- in HeLa cells. I recommend this manuscript for consideration for publication in Molecules with minor revision.

OUR RESPONSE: We thank Referee 1 very much for his/her valuable comments and recommendation. We have addressed these points as follows.

  1. Some very recent related papers are advised to be cited.

OUR RESPONSE: We thank Referee 1 for this suggestion. Some recent related papers had been added as refs. [22,23]. Please see the track-change version.

  1. Why is 90% EtOH added to the optical responses?

OUR RESPONSE: We thank Referee 1 for this comment. As shown in the figure below, SWJT-9 for hypochlorite anions works best in 90% EtOH.

Figure A. Fluorescence spectra of SWJT-9 (10.0 µM) and SWJT-9 + ClO-(2.0 mM) in different ratios of PBS and ethanol (lem = 667 nm).

3.The manuscript claimed "But these chemodosimeters for hypochlorite anion were not near-infrared probe", but as far as I know,there are some near infrared fluorescent probes for hypochlorite anion using a diaminomaleonitrile group as the recognition site. For example, Spectrochimica Acta Part A: Molecular and Biomolecular Spectroscopy 2019,219,509-516.The innovation of probe SWJT-9 needs to be further elaborated.

OUR RESPONSE: We thank Referee 1 for this comment. The overlooked probes have been added in Table S1, and cited in the main text (refs. 41-46). Although the probe mentioned by this referee is a near-infrared one (ref. 41), its Stokes shift was not large. The corresponding discussion has been revised, please see the track-change version.

  1. It is recommended to add some other reactive oxygen species to Figure 2. In addition, it is better to add error bars in Figure 2.

OUR RESPONSE: We thank Referee 1 for this suggestion. Some other reactive oxygen species and some ions have been added to Figure 2. And the error bars have also been added in Figure 2. Please see the track-change version.

  1. Why is it necessary to add such high levels of hypochlorous acid (2 mM) to the optical reaction and the cells? Does such a high measure of hypochlorite anion lead to apoptosis and thus affect the biological application of the probe.

OUR RESPONSE: We thank Referee 1 for this comment. I 'm sorry, this is a typo. Indeed, the concentration of hypochlorous anions was 200.0 μM, which has been corrected. Please see the track-change version.

Reviewer 2 Report

The manuscript by Liu et al. describes the synthesis of a near-infrared fluorescent (NIR) probe specifically designed to detect hypochlorite anion (ClO-). The authors highlight the differences to existing chemical probes that detect ClO- and indicate the advantages of this novel probe, namely working in the NIR spectrum and significant Stokes shift. The experiments and methodologies to evaluate the properties of the novel probe (detection mechanism, detection limit, detection velocity, and selectivity are appropriate. The authors also performed an in vitro experiment in HeLa cells to address potential probe toxicity and evaluate a potential application for detecting ClO- in biological systems.

The manuscript is interesting. The figures, schemes, and tables (including the supplementary materials) provide relevant support information. Nevertheless, the manuscript presents some issues, which we indicate next, and the authors should address that.

Major:
- The manuscript sections (introduction, etc.) are, overall, excessively synthetic. For instance, the material and methods, albeit the information provided in the supplementary materials, are insufficient to replicate the experiments independently. The authors should provide further detail on the reagents (purity) and the in vitro experiments (conditions for incubation, details for toxicity assays). Also, the Conclusions section, in the face of the discussion section provided, is not sufficient. For instance, authors should further discuss potential applications, limitations to this probe, etc.
- English should be revised, which is particularly necessary for the introduction section making it difficult to understand the point the authors are raising (line 32 “fluorescence probe has been…”; line 38 “…were not near-infrared probe…”; line 46 “… had been…”. There are other lesser errors and typos throughout the text (line 58; line 100; lines 135 and 136; line 152; line 154).
- Figure 1c shows the different fluorescent intensities for a range of concentrations of ClO-. The authors must indicate not a range (0-2000uM) but the specific concentrations for each color. Also, clarify and discuss why at the highest concentration (2000uM), the fluorescent intensity in figure 1(c) is ~425 whistle in figure 1(d), for the same concentration, it plateaus at ~320; in S4, it is ~500. Does this hamper the ability to use the probe as a quantification method?

Minor:
-line 26: references 8 is quite old for such a strong statement; remove or add another more recent one.
- line 71; why insert reference 49 here; these results are from the current manuscript.
- lines 78-79; clarify the meaning of “the body”; is it blood, inside cells? Insert the concentration of hypochlorite anion produced “in the body” here; also review the reference indicated as it does not refer to the concentration of ClO- “in the body.”
- line 80; this sentence deserves a reference for citation.
- figure 2: the figure would benefit from inserting the name of the analytes directly in the X-axis instead of numbers and inserting the corresponding names in the figure legend.
- figure S6: what is the concentration of SWJT-9 and the time of reaction?

Author Response

Reply to Referee 2:

Comments to the Author

The manuscript by Liu et al. describes the synthesis of a near-infrared fluorescent (NIR) probe specifically designed to detect hypochlorite anion (ClO-). The authors highlight the differences to existing chemical probes that detect ClO- and indicate the advantages of this novel probe, namely working in the NIR spectrum and significant Stokes shift. The experiments and methodologies to evaluate the properties of the novel probe (detection mechanism, detection limit, detection velocity, and selectivity are appropriate. The authors also performed an in vitro experiment in HeLa cells to address potential probe toxicity and evaluate a potential application for detecting ClO- in biological systems.

The manuscript is interesting. The figures, schemes, and tables (including the supplementary materials) provide relevant support information. Nevertheless, the manuscript presents some issues, which we indicate next, and the authors should address that.

OUR RESPONSE: We thank Referee 2 very much for his/her valuable comments and recommendation. We have addressed these points as follows.

  1. The manuscript sections (introduction, etc.) are, overall, excessively synthetic. For instance, the material and methods, albeit the information provided in the supplementary materials, are insufficient to replicate the experiments independently. The authors should provide further detail on the reagents (purity) and the in vitro experiments (conditions for incubation, details for toxicity assays). Also, the Conclusions section, in the face of the discussion section provided, is not sufficient. For instance, authors should further discuss potential applications, limitations to this probe, etc.

OUR RESPONSE: We thank Referee 2 for this comment and suggestion. The details of vitro experiments have been added in SI. The details of reagents were in the first paragraph in the section of general method in SI. And the conclusion section has been revised as well. Please see the track-change version.

  1. English should be revised, which is particularly necessary for the introduction section making it difficult to understand the point the authors are raising (line 32 “fluorescence probe has been…”; line 38 “…were not near-infrared probe…”; line 46 “… had been…”. There are other lesser errors and typos throughout the text (line 58; line 100; lines 135 and 136; line 152; line 154).

OUR RESPONSE: We thank Referee 2 for this comment and suggestion. The errors have been revised, and please see the track-change version.  

  1. Figure 1c shows the different fluorescent intensities for a range of concentrations of ClO-. The authors must indicate not a range (0-2000uM) but the specific concentrations for each color. Also, clarify and discuss why at the highest concentration (2000uM), the fluorescent intensity in figure 1(c) is ~425 whistle in figure 1(d), for the same concentration, it plateaus at ~320; in S4, it is ~500. Does this hamper the ability to use the probe as a quantification method?

OUR RESPONSE: We thank Referee 2 for this comment. The corresponding concentrations have been added in Figure 1c, and please see the track-change version.

Indeed, the fluorescence intensities were not normalization in Figure 1c, 1d, S4. This is a very common phenomenon (please see refs. 41, 42, 43, 45, 57 in main text).

  1. line 26: references 8 is quite old for such a strong statement; remove or add another more recent one.

OUR RESPONSE: We thank Referee 2 for this suggestion. It has been replaced by a recent paper, and please see the track-change version.

  1. line 71; why insert reference 49 here; these results are from the current manuscript.

OUR RESPONSE: We thank Referee 2 for this comment. The reference refers to the method used for quantum yields.

  1. lines 78-79; clarify the meaning of “the body”; is it blood, inside cells? Insert the concentration of hypochlorite anion produced “in the body” here; also review the reference indicated as it does not refer to the concentration of ClO- “in the body.”

OUR RESPONSE: We thank Referee 2 for this comment. As reported by the reference [58], the hypochlorite anions are mainly produced in cells. The corresponding discussion has been revised. Please see the track-change version.

  1. line 80: this sentence deserves a reference for citation.

OUR RESPONSE: We thank Referee 2 for this suggestion. The corresponding reference has been cited as ref. [59], please sees the track-change version.

  1. figure 2: the figure would benefit from inserting the name of the analytes directly in the X-axis instead of numbers and inserting the corresponding names in the figure legend.

OUR RESPONSE: We thank Referee 2 for this suggestion. The names of analytes have been added in the X-axis of Figure 2. Please see the track-change version.

  1. 9. figure S6: what is the concentration of SWJT-9 and the time of reaction?

OUR RESPONSE: We thank Referee 2 for this comment. The data in Figure S6 is derived from Figure 1(d), so the concentration of SWJT-9 is 10.0 mM and the reaction time is 2.0 minutes.

Reviewer 3 Report

In this study, Liu et al. designed a near-infrared fluorescent turn-on probe for hypochlorite detection. The data provided in this paper are consistent with the conclusion that the probe can be turned on by ClO-, but not enough to support the claim that this probe is selective for ClO- and can sensitively detect ClO- in cells. Overall it’s not convincing the probe presented in this work can be a useful probe for ClO- detection and some of the experiments can be better designed.

1.       In Figure 1, why are the spectra measured in 9:1 EtOH:PBS? Is it because of the poor solubility of the probe?

2.       The immine structure in the probe can be inherently unstable. The authors should provide stability data of the probe in buffer and in lysate.

3.       For the activation of the probe the authors should provide HPLC data as well.

4.       In Figure 2, the authors should include more analytes including singlet oxygen, superoxide and other common metal ions.

5.       In Figure 5, the authors should include scale bar in the fluorescence images and provide flow cytometry data for more quantitative analysis.

6.       In Figure 5, HeLa cells were treated with 2mM of ClO-, which seems awfully high when the authors claimed in their in vitro experiment the detection limit is in the nanomolar range. Are the cells even alive when treated with 2mM of ClO-?

Author Response

Reply to Referee 3:

Comments to the Author

In this study, Liu et al. designed a near-infrared fluorescent turn-on probe for hypochlorite detection. The data provided in this paper are consistent with the conclusion that the probe can be turned on by ClO-, but not enough to support the claim that this probe is selective for ClO-and can sensitively detect ClO- in cells. Overall it’s not convincing the probe presented in this work can be a useful probe for ClO- detection and some of the experiments can be better designed.

OUR RESPONSE: We thank Referee 3 very much for his/her valuable comments. We have addressed these points as follows.

  1. why are the spectra measured in 9:1 EtOH:PBS? Is it because of the poor solubility of the probe?

OUR RESPONSE: We thank Referee 3 for this comment. As shown in the figure below, SWJT-9 for hypochlorite anions works best in 90% EtOH.

Figure A. Fluorescence spectra of SWJT-9 (10.0 µM) and SWJT-9 + ClO-(2.0 mM) in different Ratios of PBS and ethanol (lem=667 nm).

  1. The imine structure in the probe can be inherently unstable. The authors should provide stability data of the probe in buffer and in lysate.

OUR RESPONSE: We thank Referee 3 for this comment and suggestion. The stability of the probe and probe + ClO‒ have been measured, please see the flowing figure. These results showed that the stability of probe was good in EtOH/PBS buffer solution.

Figure B. Time scan fluorescence at 667 nm for SWJT-9 (10.0 µM) and and SWJT-9 + ClO- (2.0 mM) in EtOH–PBS (9/1, v/v, pH 7.4) buffer solution (lex=550 nm).

  1. For the activation of the probe the authors should provide HPLC data as well.

OUR RESPONSE: We thank Referee 3 for this suggestion. We think that HPLC data is not necessary to this work.

  1. In Figure 2, the authors should include more analytes including singlet oxygen, superoxide and other common metal ions.

OUR RESPONSE: We thank Referee 1 for this suggestion. Some other reactive oxygen species and some ions have been added to Figure 2 in the main text. Please see the track-change version.

  1. In Figure 5, the authors should include scale bar in the fluorescence images and provide flow cytometry data for more quantitative analysis.

OUR RESPONSE: We thank Referee 3 for this suggestion. The scale bar in the fluorescence images have been added in Figure 5. Please see the revised manuscript with track-change. But the flow cytometry data is not available in our laboratory now.

  1. In Figure 5, HeLa cells were treated with 2mM of ClO-, which seems awfully high when the authors claimed in their in vitro experiment the detection limit is in the nanomolar range. Are the cells even alive when treated with 2mM of ClO-?

OUR RESPONSE: We thank Referee 1 for this comment. I 'm sorry, this is a typo. Indeed, the concentration of hypochlorous anions was 200.0 μM, which was corrected. Please see the track-change version.

Reviewer 4 Report

Comments:

The manuscripts describe the design and synthesis of novel near-infrared (NIR) fluorescent probe SWJT-9 for hypochlorite anions detection. Here, the reaction site for ClO detection on dicyanoisophorone skeleton was introduced by using the diaminomaleonitrile. The UV-Vis and fluorescence emission experiments performed on the probe SWJT-9 with ClO show colorimetric and fluorescence turn-on response. Moreover, the probe selectively detected the ClO   in presence of other anions and reactive oxygen species indicating the good anti-interference property. The reaction mechanism of SWJT-9 with ClO was supported with 1H NMR and mass spectral results. The change in fluorescence properties of the probe was further supported with theoretical calculations and successfully tested for the detection of ClO in HeLa cells. The present manuscript looks in detail and the supporting Information contains all the details of the performed analyses. I am recommending this manuscript to publish in Molecules journal.

Before accepting this manuscript, I would recommend/suggest authors to correct the following typographical and grammatical errors.

i)                   In lines 24, 25, 29, 31, 34, 38, 42 and so on replacing of “ion” and “anion” words with “ions” and anions” will give more appropriate meaning to the sentences.

ii)                 In line 32, replacement of “fluorescence probe” word with “fluorescence probing” or “fluorescence probing technique/method” will be more appropriate.

iii)               In line 34, It would be more appropriate to replace “probe” and “anion” words with “probes” and “anions”.

iv)               In line 70 replacement of “for fluorescence spectrum” with “in the fluorescence spectrum” will be more accurate.

v)                  In line 72, it would be better to continue the sentences by replacing “[49]. And” with “[49], and” (also in lines 173-174).

vi)               In Scheme 1, the letter P in “Piperidine” on first arrow should be lowercase.

vii)             In line 74, “The concentration titration” could change to “The fluorescence titration”.

viii)           In line 76, SWJT-9 was mistakenly written instead of ClO-, “SWJT-9 concentration” should change to “ClO− concentration”. (see figure 1c)

ix)               In line 86, Figure 1 “FL.Intensity” could be written as  FL.Intensity (a.u). and in 1a, “wavelength” the W letter should be capitalized.

x)                  In lines 89 and 91, replacement of “Inset: images of SWJT-9 and SWJT-9 + ClO−” with “Inset: images of SWJT-9 (left) and SWJT-9 + ClO (right)” would better clarify to readers which image is which.

xi)               In line 118, replacement of “the peak at m/z 349.3” with “the peak at m/z 349.3 of SWJT-9 + ClO which corresponds to 3” will clarify/justify the author’s statement.

xii)             In line 132, changing of Figure 4 label to “Optimized structures and molecular orbital plots of SWJT-9 and Compound 3” will be more appropriate than the present label “Molecular structure and optimized orbitals of SWJT-9 and Compound 3”. Also, showing the HOMO and LUMO values with the band gap in the molecular orbital figure will be better for the reader’s understanding of the absorption spectra and justification of the redshift shown in Figure 1a.

xiii)           In line 138, “then ClO– was added for 20 minutes” implies that ClO was added over the course of 20 minutes (slow addition). It might be better to write “then incubated with ClO– for another 20 min”. Also, check the same in description of Figure 5 (lines 144-145). Unless the ClO– was slowly added for the 20-minute duration. As written, it is ambiguous to the reader.

xiv)           In line 190, no authors are given for reference. DOI: 10.1146/annurev-pathol-020712-164023

xv)              In line 195, refence 4, journal abbreviation is not correct. DOI: 10.2527/jas.2007-0620.

Author Response

Reply to Referee 4:

Comments to the Author

The manuscripts describe the design and synthesis of novel near-infrared (NIR) fluorescent probe SWJT-9 for hypochlorite anions detection. Here, the reaction site for ClO detection on dicyanoisophorone skeleton was introduced by using the diaminomaleonitrile. The UV-Vis and fluorescence emission experiments performed on the probe SWJT-9 with ClO show colorimetric and fluorescence turn-on response. Moreover, the probe selectively detected the ClO   in presence of other anions and reactive oxygen species indicating the good anti-interference property. The reaction mechanism of SWJT-9 with ClO was supported with 1H NMR and mass spectral results. The change in fluorescence properties of the probe was further supported with theoretical calculations and successfully tested for the detection of ClO in HeLa cells. The present manuscript looks in detail and the supporting Information contains all the details of the performed analyses. I am recommending this manuscript to publish in Molecules journal.

OUR RESPONSE: We thank Referee 4 very much for his/her valuable comments and recommendation. We have addressed these points as follows.

  1. In lines 24, 25, 29, 31, 34, 38, 42 and so on replacing of “ion” and “anion” words with “ions” and anions” will give more appropriate meaning to the sentences.
  2. In line 32, replacement of “fluorescence probe” word with “fluorescence probing” or “fluorescence probing technique/method” will be more appropriate.
  3. In line 34, It would be more appropriate to replace “probe” and “anion” words with “probes” and “anions”.
  4. In line 70 replacement of “for fluorescence spectrum” with “in the fluorescence spectrum” will be more accurate.
  5. In line 72, it would be better to continue the sentences by replacing “[49]. And” with “[49], and” (also in lines 173-174).
  6. In Scheme 1, the letter P in “Piperidine” on first arrow should be lowercase.
  7. In line 74, “The concentration titration” could change to “The fluorescence titration”.
  8. In line 76, SWJT-9 was mistakenly written instead of ClO-, “SWJT-9 concentration” should change to “ClO− concentration”. (see figure 1c)
  9. In line 86, Figure 1 “FL.Intensity” could be written as FL.Intensity (a.u). and in 1a, “wavelength” the W letter should be capitalized.
  10. In lines 89 and 91, replacement of “Inset: images of SWJT-9 and SWJT-9 + ClO−” with “Inset: images of SWJT-9 (left) and SWJT-9 + ClO (right)” would better clarify to readers which image is which.
  11. In line 118, replacement of “the peak at m/z 349.3” with “the peak at m/z 349.3 of SWJT-9 + ClO which corresponds to 3” will clarify/justify the author’s statement.
  12. In line 132, changing of Figure 4 label to “Optimized structures and molecular orbital plots of SWJT-9 and Compound 3” will be more appropriate than the present label “Molecular structure and optimized orbitals of SWJT-9 and Compound 3”. Also, showing the HOMO and LUMO values with the band gap in the molecular orbital figure will be better for the reader’s understanding of the absorption spectra and justification of the redshift shown in Figure 1a.
  13. In line 138, “then ClO– was added for 20 minutes” implies that ClO was added over the course of 20 minutes (slow addition). It might be better to write “then incubated with ClO– for another 20 min”. Also, check the same in description of Figure 5 (lines 144-145). Unless the ClO– was slowly added for the 20-minute duration. As written, it is ambiguous to the reader.
  14. In line 190, no authors are given for reference. DOI: 10.1146/annurev-pathol-020712-164023
  15. In line 195, refence 4, journal abbreviation is not correct. DOI: 10.2527/jas.2007-0620.

OUR RESPONSE: We thank Referee 4 for this comment and suggestion. We have try our best to revise these errors in the manuscript. Please see the revised manuscript with track-change.

Round 2

Reviewer 3 Report

The authors have addressed most of my previous comments to my satisfactory except the following:

1.       The authors did not provide an explanation to why a 9:1 EtOH/PBS was used. Simply by saying this is optimum is not enough. Was it because the probe has poor solubility or because the probe aggregates in aqueous buffer?

Author Response

Reply to Referee 3:

Comments and Suggestions for Authors

The authors have addressed most of my previous comments to my satisfactory except the following:

  1. The authors did not provide an explanation to why a 9:1 EtOH/PBS was used. Simply by saying this is optimum is not enough. Was it because the probe has poor solubility or because the probe aggregates in aqueous buffer?

OUR RESPONSE: We thank Referee 3 for this comment. As referee said, the solubility of probe is not so good and it will aggregates in high water phase. The corresponding discussion was added in the main text. Please see the track-change version.
